Subject Areas:
developmental biology/computational biology

Keywords:
cell movements, pattern formation, evolution of complex systems, evo-inspired engineering

Authors for correspondence:
Berta Verd
e-mail: berta.verdfernandez@zoo.ox.ac.uk
Benjamin Steventon
e-mail: bjs57@cam.ac.uk

# The unappreciated generative role of cell movements in pattern formation

Timothy Fulton[1,2], Berta Verd[1,3] and Benjamin Steventon[1]

[1]Department of Genetics, University of Cambridge, Cambridge, UK
[2]School of Biological and Behavioural Sciences, Queen Mary University of London, London, UK
[3]Department of Zoology, University of Oxford, Oxford, UK

 BV, 0000-0001-9835-009X; BS, 0000-0001-7838-839X

The mechanisms underpinning the formation of patterned cellular landscapes has been the subject of extensive study as a fundamental problem of developmental biology. In most cases, attention has been given to situations in which cell movements are negligible, allowing researchers to focus on the cell-extrinsic signalling mechanisms, and intrinsic gene regulatory interactions that lead to pattern emergence at the tissue level. However, in many scenarios during development, cells rapidly change their neighbour relationships in order to drive tissue morphogenesis, while also undergoing patterning. To draw attention to the ubiquity of this problem and propose methodologies that will accommodate morphogenesis into the study of pattern formation, we review the current approaches to studying pattern formation in both static and motile cellular environments. We then consider how the cell movements themselves may contribute to the generation of pattern, rather than hinder it, with both a species specific and evolutionary viewpoint.

## 1. Introduction

Embryonic development is a process which happens across scales, requiring temporal and spatial coordination across multiple levels of organization. At the smallest scale, individual cells must make numerous cell fate decisions as they acquire their final fates. This must be organized in such a way that coordination between cells produces patterns that, at the multi-cellular and tissue scale have organization and coherence.

Ultimately, the well-orchestrated development of multiple tissues lead to a reproducible development of embryos into their final adult form. A key question in developmental biology therefore relates to how information is propagated across these scales, both in terms of developmental emergence (i.e. how

alterations within cells result in the emergence of tissue patterning and morphogenesis), and downward causation (i.e. how information is relayed downwards from alterations at the organismal and multi-tissue level to the regulation of dynamic individual cellular processes). How such multi-level interactions lead to the timing of developmental processes has been reviewed recently [1]. Here, we focus on how patterning (the generation of discernible patterns of gene expression) and morphogenesis (tissue shaping) act together.

Cellular movements are a critical feature of embryogenesis, as it is the driving force of morphogenesis, which shapes the embryo into its final form. By regularly reorganizing cells and tissues in space, this results in the frequent rearrangement and repositioning of signalling centres which in turn can function either to further develop the pattern being produced by exposing cells differently to signalling, or to disrupt it by blurring the boundaries between domains of gene expression. As the pattern of gene expression within a tissue is a function of both the dynamics of gene expression intrinsic to the cells and the temporal exposure to extrinsic signals, cell movements probably act as an important additional component in the regulation of pattern emergence. In addition to this, mechanochemical signals can also impact the regulation of gene expression by triggering the activity signalling cascades in response to an altered mechanical environment [1]. Given these observations, we propose that by the explicit incorporation of cell movement into our understanding of pattern emergence, new mechanisms of this fundamental problem in biology are likely to emerge. This will build on our current understanding of how fate decisions in individual cells result in tissue-level molecular patterns that have largely been derived from studies in tissues with limited cell movement. Such studies have clearly demonstrated how, for example, morphogen gradients are able to inform cells of their position within a tissue and how cell fates are resolved as a result [2–6].

The aim of this review is to draw attention to the largely unappreciated role of cell movements in pattern formation. To this end, we will briefly review some of the landmark studies of pattern formation in tissues with limited cell movement to emphasize how in such systems pattern formation is an emergent property of signalling and gene regulatory networks (GRNs) alone, and briefly consider how research in these systems has shaped our current notions of patterning precision. We then move on to review pattern formation in developing tissues with extensive cell movements to propose that in such cases, pattern formation is no longer an emergent property of signalling and GRNs alone, but of signalling, GRNs and cell movements. We go further to suggest that cell movements themselves may play an important and often overlooked generative role in patterning, rather than being merely a source of noise to be buffered. Finally, we will consider how this broader conceptualization of the drivers of pattern formation might help us understand how developmental patterning might evolve by modifying not only signalling environments and GRN interactions across phylogeny, but also the geometry of body plans, with different cell and tissue morphogeneses.

## 2. Pattern formation when cell movements are absent or negligible

Two major models have been proposed to explain how patterns of gene expression form across a field of static cells [7]: positional information and reaction–diffusion. Lewis Wolpert developed the idea of positional information to help understand how cells that maintained their relative position (coordinates) within a tissue might be able to decipher their identity through the interpretation of concentration gradients of signalling factors [8] which were later called 'morphogens' [9]. This conceptualization of the mechanisms underlying pattern formation has proven incredibly useful to understand various developmental patterning systems. Of these, perhaps the most notable is anterior–posterior patterning of the early Drosophila embryo, where various signalling gradients—most famously, the Bicoid protein gradient—were identified and shown to play important roles positioning the downstream gap and pair-rule gene boundaries (reviewed in [5,6]). These works demonstrated that morphogen concentration gradients are indeed able to pre-pattern major body axes during embryogenesis. As a result, positional information was adopted by the field as a readily applicable general mechanism for pattern formation, and since, little attention has been paid to the extent to which its applicability depends on cells having constant coordinates within the tissue being patterned.

The Drosophila blastoderm is quite an unusual cellular environment [5]. During the early stages of gap and pair rule gene patterning, the blastoderm is in a syncytial stage which becomes cellularized as the pair-rule and segment polarity patterns become established shortly before the onset of gastrulation. During this time the nuclei, then cells, divide but do not mix or move, hence maintaining their relative coordinates within the tissue. This unique feature makes the blastoderm

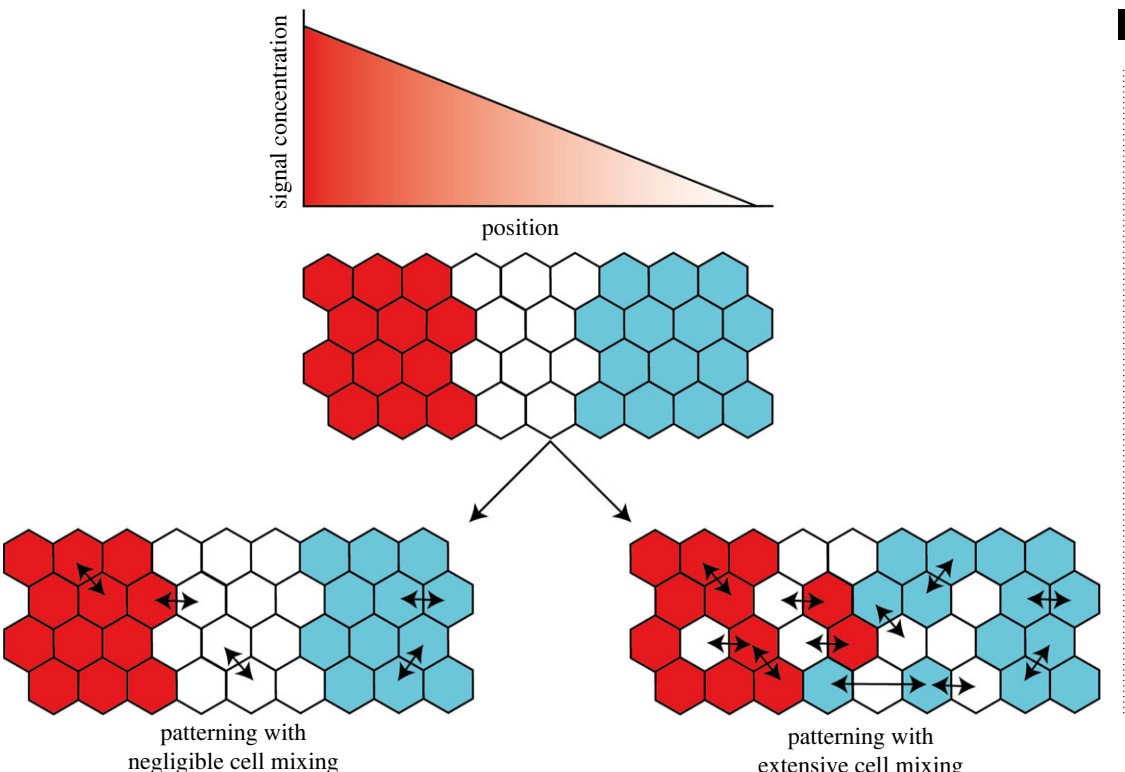

**Figure 1.** Cell mixing scrambles patterns produced via positional information when patterning occurs over longer timescales than cell movement. In situations where spatial coordinates are given by morphogen gradients, a pattern can be produced. This pattern, however, will be rapidly dispersed with the introduction of cell movements which occur on a sufficiently different timescale to patterning.

particularly amenable to positional information-based mechanisms and explanations, while also highlighting how stringent this requisite can be. If cells do change their positions, and therefore their relative coordinates within the tissue as the pattern emerges, it becomes much less intuitive when and how they might be inferring their position from morphogen gradients. It is therefore difficult to imagine how a positional information-based patterning mechanisms might work in general. However, cell movements do not have to be completely absent for positional information to apply, at least intuitively, they just need to be negligible. Cell movements will be negligible if the cells are moving but not effectively changing their coordinates along the axis being patterned or if the timescales of cell movements and pattern formation are different enough such that they can be effectively uncoupled and addressed separately (figure 1). To assess whether or not cell movements will have a significant impact on pattern formation will require the integration of cell tracking data within models of this complex process as we will describe below.

In cases where cell movements are absent or at least negligible, pattern can be considered an emergent property of GRNs and signalling alone. The field has developed efficient methodologies that allow us to reverse-engineer data-driven mathematical models representing the GRNs driving pattern formation in measured signalling scenarios. These approaches have been very successful at explaining pattern formation and in particular the contribution of morphogens via positional information in systems such as the fore-mentioned anterior-posterior patterning in the Drosophila blastoderm, pattern formation in the vertebrate neural tube and the limb bud [3,4,10] among many others (reviewed in [11]).

All of these approaches implicitly depend on being able to assume that the quantification of spatio-temporal gene expression patterns across a tissue reflects the gene expression dynamics of the single cells at every position. However, this will hold true if cells retain (or can at least be assumed to retain) the same coordinates throughout the patterning process. If cells move around and dynamically update their spatial coordinates relative to the tissue over time, tissue-level quantification of gene expression will poorly represent gene expression dynamics in single cells, making it impossible to infer the GRNs driving those dynamics with any reasonable accuracy. For the sake of simplicity and progress, as a field we have disproportionately studied patterning processes where such assumptions hold true, learning a great amount about such systems as a result, while in systems where the timescales of cell

movements and pattern formation cannot be separated, our understanding of the mechanisms driving pattern formation remains rudimentary at best.

Positional information is not the only conceptual model of pattern formation that struggles when cell movements are considered explicitly. Reaction–diffusion systems too have been formulated with the implicit assumptions that the cells that compose the tissue being patterned remain static. This model was initially proposed by the mathematician Alan Turing, and shows how external inputs are not required for the formation of pattern, and that pattern (famously spots and stripes) can spontaneously emerge from noise in an initially homogeneous field of static cells [7,9,12]. The mechanism he proposed relied on two chemical species (which could be mRNA molecules), a repressor and an activator, with different diffusion rates acting across a static field of cells, and has since been extended to include more factors. When cells are static, the different diffusion rates of the activator and the inhibitor suffice to bring about the spontaneous formation of pattern. However, if the cells from which these factors emanate are moving around at speeds of comparable scale to the diffusion rates, the activator and inhibitor distributions across the tissues will be affected. Simulations studies would be required to begin to understand the extent and nature of the disruption to the pattern, and how it depends on the scales of diffusion and cell movements.

Some of our ideas of patterning precision and reproducibility are deeply linked to the field's tendency to simplify cell movements out of the patterning equation. Patterning precision has been interpreted in a number of different ways within the developmental biology literature. Here, we will briefly discuss the implications of considering cell movements in terms of the reproducibility of gene expression concentrations at corresponding positions in multiple embryos [13].

How pattern precision between individual embryos remains is an important question, and is made further difficult to understand in tissues where large numbers of cells are rearranging. To achieve pattern robustness, either cells must undergo highly stereotypical movements between one embryo and the next, or they must be able to regulate their gene expression state such that robust gene expression patterns can emerge. It seems that the latter is more likely, as it is hard to imagine how the precise rates and directionality of cell rearrangement can be constrained to a high degree for large numbers of cells. In this scenario, the relative timescales of cell rearrangement, morphogen sensing and gene regulation all become important. Cells must be able to continually update their gene expression state as they move closer or further away from a signal source, yet also possess their own intrinsic timing of signal response. To probe this regulative ability further, we must tackle how developmental patterning systems can adapt to experimental perturbations in the tissue geometry, extrinsic mechanical forces, or other perturbations that will impact cell movements and tissue morphogenesis. Simultaneously assessing relative timescales of movement, signalling and gene expression in such a context is technically challenging. However, it could be hugely beneficial to uncover the developmental mechanisms of pattern regulation and robustness.

Exactly how boundaries between different cell types form has been a long-considered question, particularly well studied using the nervous system. In the zebrafish neural tube, differently expressed cadherins resulting in differential adhesion [14], in combination with the gradient of Shh results in sorting of like cells [15]. Differential adhesion is also observed operating within the formation of the distinct domains which make up the hindbrain rhombomeres [16–18]. Here cells are actively sorted into domains using cell–cell signalling mediated via Eph-ephrin [19] which results in differential adhesion between like- and non-like cell types [20]. In addition to this cell sorting based on differential expression of cell adhesion molecules, mutually exclusive gene expression patterns ensure that the boundaries between domains are sharp, with no co-expression of markers from adjacent domains within single cells [21]. Together, these examples demonstrate the process of active cell sorting, where cells are organized into domains by the activity of distinct gene expression profiles within each cell. This, however, contrasts with the idea of extensive cell mixing where the movement of cells occurs over long ranges, with the expected outcome of mixing the observed patterns rather than refining them. To highlight this problem, we will now consider how cells dynamically update their gene expression to maintain coherent domains of gene expression at the tissue level.

## 3. Pattern formation during extensive cell rearrangement

Despite the examples given previously, pattern formation in developing embryos also happens during periods of simultaneous extensive cell rearrangement, where cells routinely change places with their neighbours in a way which would be expected to disrupt boundary formation, rather than refine such

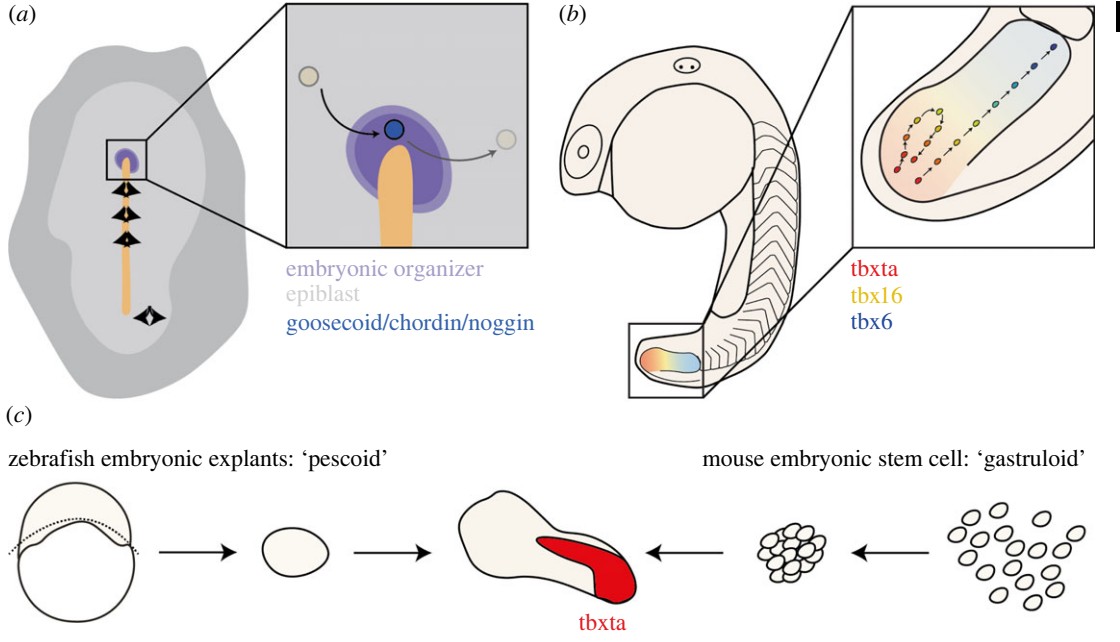

**Figure 2.** A series of examples of pattern formation during extensive cell mixing and rearrangement: (*a*) the chicken organizer, (*b*) the zebrafish presomitic mesoderm and (*c*) embryonic self-assembly *in vitro* of zebrafish pescoids, made from whole embryonic explants taken prior to gastrulation, and mouse gastruloids, made from embryonic stem cells. Human gastruloids are made in the same way as mouse gastruloids.

boundary. In order to retain some degree of pattern regulation, it is therefore essential that a cell is able to dynamically update its gene expression state to keep pace with the updating of the cell's position within the tissue. If this was not the case, over time the patterns formed across the tissue would progressively diffuse and become lost. The process of cellular differentiation and cellular movement must therefore be coupled to generate domains of gene expression forming at the tissue level. This idea will be illustrated using three examples where patterns are produced in the context of cell rearrangements: organizer formation and maintenance, vertebrate somitogenesis and self-assembly of embryonic-like structures *in vitro*.

One major process during gastrulation (figure 2*a*) is the formation of the organizer; a group of cells within the embryo that are able to determine the fate and morphogenesis of cells around them [22]. The Node (the organizer in avian species) is responsible for significant patterning events, including left–right patterning [23–25] and anterior–posterior patterning of the neural tissue [26,27]. These varied roles are achieved by functioning as a source of multiple signalling cues which refine the broad early patterns [22]. It was noted in the study of the organizer in chicken embryos that cells routinely entered and exited the region of the embryo defined as the Node, and in doing so are required to upregulate or downregulate organizer genes such as *goosecoid*, *noggin* and *chordin* [28]. This therefore suggests that while the Node itself persists during development, its components are transient, with cells constantly entering and exiting it. Further to this, total removal of the organizer and replacement with cells from other regions of the epiblast results in reformation of the organizer via signalling from a 'Node Inducing Centre' in the primitive streak [28–31]. These results together demonstrate the determination of cells within the organizer region to form and maintain organizer identity is given by the position of cells within the embryo rather than by the specific cells themselves [22]. This also demonstrates the region of gene expression that gives rise to organizer function must be maintained even though cells are continually moving in from the surrounding epiblast. Importantly, recent work has demonstrated how the Node functions as a stem cell niche to confer stem cell identity to population of resident stem cells, thereby continually specifying epiblast cells to an organizer state as they enter the niche [32]. How the dynamics of cell fate assignment, cell movement and signal response are tuned at the single-cell level remains an essential question for future study.

Such robustness of pattern formation to cell mixing is also observed later in development, during the process of somitogenesis (figure 2*b*). This process couples pattern formation with significant morphological change, as the somites (segmented blocks of mesoderm) bud off sequentially from the anterior presomitic mesoderm while the embryonic axis continues to elongate. In addition to these

large morphological changes in tissue structure, individual cells within the unsegmented presomitic mesoderm also change in cell movements and behaviour in a species-specific manner. Recent studies in the zebrafish embryo, which investigate the movements of cells across the tissue, identified a stark transition between high levels of cell motility, with a high number of neighbour exchanges in the posterior of the unsegmented presomitic mesoderm and the more anterior presomitic mesoderm where cell rearrangement was reduced and the overall tissue more rigid [33–37]. In addition to this, local events which define the positioning of the somite itself as a response to physical changes in tissue properties have been identified [38].

On top of the movements within the progenitor region of the pre-somitic mesoderm (PSM), patterns of gene expression form, with up to two somites in the zebrafish being prepatterned in the anterior unsegmented presomitic mesoderm prior to forming a true segmented morphological somite [39,40]. The Clock and Wavefront model [41] has been proposed as a mechanism to explain the temporal and spatial formation of somites during axial elongation. Through the interaction of segmentation clock gene oscillations (Her1/Her7 oscillations in zebrafish) [42] and an anterior to posterior receding threshold of Wnt and FGF signalling [43,44] alongside anteriorly originating retinoic acid [45], the positioning of the somites at the anterior of the presomitic mesoderm is achieved; reviewed in [46]. The oscillations of Her1, which in the embryo are synchronized by Notch signalling, have also been demonstrated to be cell autonomous within *in vitro* cultures of individual presomitic mesodermal cells [47,48]. This cell autonomy, however, requires regulating once these cells are assembled into a tissue in order to produce coherent Her1 waves of expression. It has been demonstrated that the synchronization of the oscillations is, in part, brought about through Delta-Notch signalling via direct cell–cell contact, with mathematical models suggesting that cells rapidly exchanging neighbours is responsible for the synchronization of the oscillations of Her1 [49]. This provides an example of how the rearrangement of cells plays a generative role in the emergence of patterning at the tissue level.

A stark example of the ability of embryonic cells to form robust patterns despite cell mixing is demonstrated in the annual killifish, *Nothobranchius furzeri*. These fish undergo the highly unusual process of cell cycle arrest [50] and cell delamination during epiboly. During delamination, the embryonic cells dissociate from one another and migrate towards the opposite pole of the egg, in doing so entirely randomizing their position within the embryo proper prior to reassociation and the initiation of gastrulation, expression of Brachyury and formation of the embryonic organizer [51–53]. Despite this extensive cell rearrangement, where every cell's starting position is randomized, the embryo is still able to undergo a process of self-assembly, driven by Nodal signalling [54] acting as a very short-range morphogen to reform the embryo at only a single position on the yolk, and initiate germ layer patterning. This reformed embryo is then able continue through development to form a fully formed adult, thereby displaying that the early embryonic cells are able to self-assemble, organize and pattern successfully.

This process of self-assembly, where cells aggregate and establish a pattern is not unique to the annual killifish and has also been observed experimentally after dissociating whole embryonic explants of Atlantic killifish [55], zebrafish [56,57] and cavefish [58] embryos. These explants have recently been termed *Pescoids* (figure 2c). Taken prior to gastrulation and the mid-blastula transition, zebrafish pescoids have been demonstrated to undergo substantial cell mixing prior to elongation, with small groups of cells deriving clones which disperse across the entire explant by 5 h post culture [56]. Tracking small clusters of labelled cells within the blastomere margin of pescoids revealed only local cell rearrangement over the first 2 h post culture [57], but appears to occur later as the explants become polarized with respect to germ layer markers [56]. This patterning has also been demonstrated in zebrafish animal caps which, when injected with Nodal, elongate and specify all three germ layers. Without this Nodal signal, however, such elongation and patterning is not observed [59], as is also the case in whole embryo explants where early Nodal signalling is inhibited [56,57]. Understanding how Nodal signalling coordinates cell movements and pattern formation in the context of cell assembly is an exciting question for the field.

Pescoids have been used to demonstrate a direct connection between correct pattern formation and cell movements. In the zebrafish, it was observed that separation of a domain of high BMP signalling from a domain undergoing high Wnt/TCF signalling occurred simultaneously with the breaking of radial symmetry and formation of an elongated structure from a previously spherical cluster of cells [56,57]. By inhibiting the convergence and extension movements responsible for the formation of this elongation, the BMP and Wnt/TCF signalling domains remained overlapping and therefore patterning of the central hindbrain region of the explants, marked by two stripes of Krox20 was prohibited [56].

This finding demonstrates a mechanism where the tissue's convergence and extension movements regulate cell's exposure to sources of signalling in terms of intensity and time period, and therefore these tissue movements have a direct impact on cell fate decisions.

An additional observation from explants of the early zebrafish embryo is that polarized *tbxta* expression can still be observed when all cells are dissociated, mixed and re-aggregated again [56]. This is strongly suggestive of a degree of self-organization in early axis specification events that is supplemented during normal development by extra-embryonic positional cues. It highlights the need to determine how spatial domains of early germ layer markers can be established and maintained in the context of cell rearrangement in experimental situations, as such knowledge will probably reveal new mechanisms of developmental robustness that act in parallel to established mechanisms of axis specification. A capacity for self-organization has also been demonstrated in mammalian *in vitro* models of gastrulation using both mouse and human ES cells [60–62]. Aggregates of a small number of these cells together followed by culturing over a number of days demonstrates that these too are able to break symmetry and form the three primary germ layers with patterning along the anterior-posterior (AP), dorsal-ventral (DV) and medio-lateral (ML) axes. Together, the embryonic explant model and stem cell *in vitro* models demonstrate clear feedback from cell movements and cell organization onto pattern formation. Both models display a high level of continuous cell rearrangement during the process of pattern formation yet reproducibly form similar body plans and gene expression domains.

# 4. Approaches to model pattern formation in the context of cell rearrangement

To date, the current approaches to study dynamics of pattern formation in cells within a tissue take mathematical modelling approaches to elucidate how signals and transcription factors function in a GRN in order to create patterns. As described previously, these approaches have been very successful and demonstrate the potential power of morphogen gradients in the patterning of tissues; however, they cannot be used in general to understand pattern formation in the presence of extensive cell mixing. Computational modelling is proving a powerful tool to probe complexities associated with both morphogenesis and patterning during development. Here, we will briefly review several scenarios where both patterning and cell movement have been tackled together, though refer the reader to a more in depth review on the subject [63].

In cases where cell numbers are low enough to obtain quantitative information about cell shape changes, it has been possible to accurately model the impact of tissue morphogenesis on morphogen gradient interpretation and patterning. Recent examples include the discovery that cell-to-cell contact area is critical for early patterning of the ascidian, *Phallusia mammillata* [64]. Furthermore, multi-dimensional modelling of plant development has revealed significant insights into the intersections of tissue morphogenesis and patterning [65]. However, the examples given in this review deal with tissues composed of hundreds to thousands of cells and therefore present a distinct problem for composite models that attempt to simultaneously predict morphogenesis and pattern formation. Therefore, we will focus on recent work that has combined dynamical systems models of GRNs with empirically derived cell tracking data to predict gene expression pattern emergence in the context of complex tissue morphogenesis.

Through the recent development on high-resolution *in toto* imaging of developing embryos using light-sheet microscopy [66,67], and the tracking of individual cells with a high degree of accuracy over a number of hours, it is now possible to consider the impact of different types of cell movements on cell fate decisions. Light-sheet imaging provides sufficient spatial and temporal resolution in order to track individual cells, without the phototoxicity associated with other microscope set-ups [68,69]. Prior to this technological advance, cell tracking tended to follow population level dynamics, demonstrating how cohorts of cells move through tissues. With advances in single-cell resolution imaging, individual movements can measured. Having this high resolution will allow for the identification of cells which seem to fall outside of the expected behaviour, such as those described previously which enter and exit the chicken embryo Node, or move posteriorly, away from the differentiation front in the zebrafish presomitic mesoderm [70].

With this improved understanding of single-cell movement dynamics, we are now able to study how cell movements, in combination with GRNs function to generate pattern (figure 3). In a recent study, such in toto cell tracking datasets have been combined with quantification of T-box gene expression and Wnt

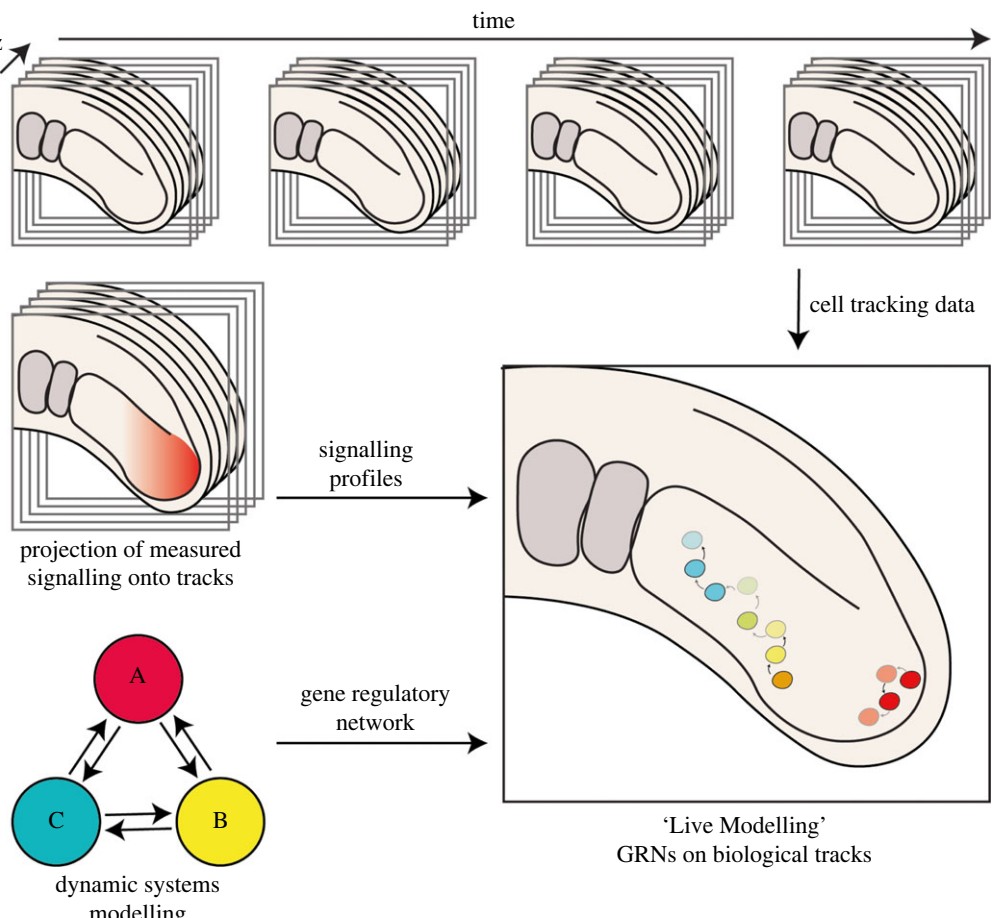

**Figure 3.** Live Modelling of pattern formation: integration of cell movements with dynamical systems modelling. Using *in toto* live imaging of developing tissues and tracking individual cells in three dimensions, accurate cell-level tracking data can be produced. Onto this, signalling profiles measured from fixed embryos are projected. A reverse-engineered dynamic systems model can be simulated onto the cell tracking data with inputs from signalling information to generate a 'Live Model' with cells simulating gene expression while undergoing real cell movements and tissue morphogenesis.

and FGF signalling activity to reverse engineering of GRNs in the context of cell movements in the zebrafish PSM [70,71]. By approximating gene expression trajectories associated with cell tracks across different regions of the tissue, it was possible to infer sets of GRN parameters sufficient to predict the dynamics of T-box gene expression changes according to the associated temporal profile of Wnt and FGF exposure for each cell [71]. To narrow down potential GRNs, the system was further challenged by explanting posterior progenitor cells and observing the dynamics of T-box expression in culture, together with the impact on Wnt and FGF signal activity [70]. Additional constraints were imposed by selecting networks that have a previously shown activation of Tbx6 by Tbx16 [72,73], and *tbx16* by FGF activity [74,75]. Finally, to test if this GRN was able to predict the emergence of T-box gene expression on all cells with the PSM, every cell was assigned a changing Wnt and FGF activity profile depending on their changing position relative to their signalling gradients. The resulting 'Live Modelling' of the T-box gene expression revealed how gene expression patterns emerge through the tuning of gene expression dynamics as a consequence of both GRN interactions and cell movement. Through such a Live Modelling approach, it may be possible to study how cells input and respond to spatial information, such as morphogen gradients, using GRNs in order to make cell fate decisions in a spatially regulated way while also moving around within space. This opens the door to tackling questions such as how GRNs are able to stably maintain domains of gene expression, such as the embryonic organizer, while also being sufficiently flexible in fate such that cells can opt for another cell identity should the cell be displaced from the expression domain.

In addition to the study of signals as morphogen gradients which inform cell states, signals are also known to regulate cell movements [35,73,74]. Mutants or the pharmacological treatment of embryos to

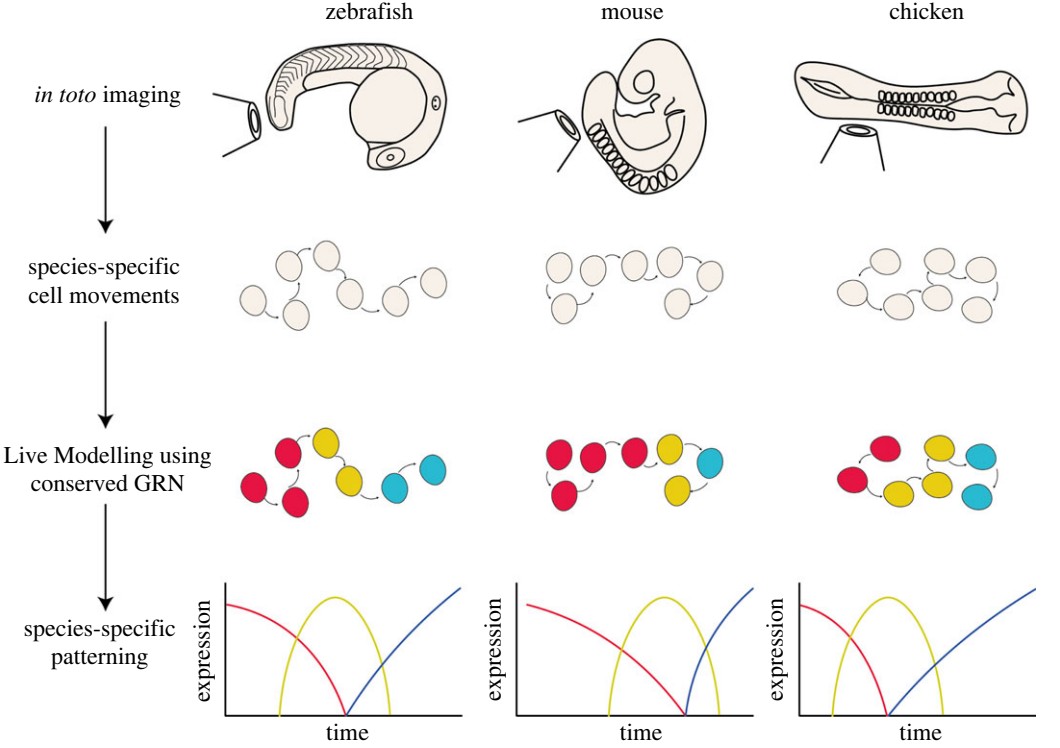

**Figure 4.** Evo-Live Modelling allows the simulation of a conserved reverse engineered gene regulatory network onto species-specific cell tracking data, resulting in predictions of species-specific patterning which can then be experimentally validated.

inhibit signalling display alterations in both the cell fate choices and also how the cells move within the field. One such example is the inhibition of FGF signalling in zebrafish resulting in both a change in mesoderm patterning [73,74], and also cell movements [35,74,76]. Uncoupling these two outcomes of a single perturbation is currently very difficult; however, a Live Modelling approach can prove useful. With live imaging, it is possible to generate tracking data from embryos with disrupted signalling in order to capture real cell movements following signal modification. The resulting signalling profiles in these embryos can then be quantified and used to simulate pattern formation in a perturbed signalling scenario which takes into account the effect of signalling on cell movements. These simulations can then be directly compared with those obtained using wild-type signalling profiles and tracks, to help disentangle the role of signalling in morphogenesis and cell fate decision making. In addition, if changes in the patterning are observed using perturbed signalling on wild-type tracks, this predicts the function of signalling on patterning. However, if different changes are predicted following the use of signalling perturbed tracks with wild-type signalling, this suggests the reverse, that the pattern is being formed as a consequence of cell movement variability. This idea could be extended to tackle the problem of GRN components which function to both regulate cell identity and also cell movement; a major obstacle in understanding the function of the T-box transcription factors during somitogenesis as mutants and morphants of the key players also display significant cell movement abnormalities [73,75,77–79].

The Live Modelling approach may be taken one step further with the consideration of how variation in cell movements have generated variation in pattern formation across evolutionary comparisons. It is probable that the core components of many cell fate decision-making GRNs are largely conserved between closely, and less closely, related species. These species, however, are likely to have differences in how the output of these GRNs manifest themselves. These differences in patterning are likely to be, in part, due to additional factors that will impact tissue morphogenesis of one species compared with another, and not only due to the strength of interactions between network nodes. Such factors might include alterations in the metabolic state of cells, tissue geometry, cell density of the structure of extracellular matrix. Using a Live Modelling approach, it will be possible to explore this by fitting and simulating GRNs on a range of tracks taken from different species and comparing the patterns produced with the pattern observed *in vivo* within each species. Such an approach would enable the exploration of a role for cell movements in generating variability in pattern formation (figure 4).

# 5. Conclusion

In summary, the question of how patterns form across tissues is one which has been considered for decades. A number of landmark proposals, such as the morphogen gradient, have been made in this time which have shaped how we think about the process of how cell fate determination happens. Using dynamical systems modelling, we have been able to accurately predict how cells behave within a morphogen gradient and have been able to predict how *in vivo* tissues will pattern using mathematical modelling. The next challenge is to consider how these dynamical systems models can be elevated further in order to consider both how cells interpret a morphogen gradient, but also how they can dynamically respond to changes in cell state as cells move around within a tissue. We propose that a combination of high-resolution *in toto* imaging, producing cell resolution tracking data, in combination with more traditional GRN modelling may rise to such a challenge. Using this approach, we may not only be able to understand how single cells dynamically update their cell state to stay coordinated with their position within the embryo, but also how signals and GRNs function together to regulate both cell movements and cell state. Lastly, we may also then be able to consider how changes in cell movements and morphogenesis have functions as tunable inputs to pattern formation between closely, and less closely related species which otherwise function using a mostly conserved GRN.

Data accessibility. This article has no additional data.
Authors' contributions. T.F.: writing—original draft, writing—review and editing; B.V.: writing—original draft, writing—review and editing; B.S.: writing—original draft, writing—review and editing.

All authors gave final approval for publication and agreed to be held accountable for the work performed therein.
Conflict of interest declaration. We declare we have no competing interest.
Funding. B.V. was supported by a Herschel Smith Postdoctoral Fellowship, University of Cambridge and Department of Zoology, University of Oxford. B.S. is supported by a Henry Dale Fellowship jointly funded by the Wellcome Trust and the Royal Society (109408/Z/15/Z) and T.F. by a scholarship from the Cambridge Trust, University of Cambridge.

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
