## [Peer Review File · Royal Society Open Science]

Review History

RSOS-211293.R0 (Original submission)

Review form: Reviewer 1

Is the manuscript scientifically sound in its present form?

No

Are the interpretations and conclusions justified by the results?

Yes

Is the language acceptable?

No

Do you have any ethical concerns with this paper?

No

Have you any concerns about statistical analyses in this paper?

No

Recommendation?

Major revision is needed (please make suggestions in comments)

Comments to the Author(s)

In their manuscript, Fulton et al review an interesting current question in developmental biology – what is the role of cell movements in pattern formation. Recent advances in imaging and mathematical modeling have allowed new insights to be gained into this question, hence the review is timely and relevant. The idea and general structure of the manuscript are well conceived. However, as I outline below, there are major issues that make the manuscript unsuitable for publication in its present form.

1. The writing quality is one of the poorest I have seen in a submitted manuscript. Grammatical errors often make it impossible to understand what the authors are trying to say. I am giving several examples below, but the text will need to be thoroughly edited.

-Run on sentences are extremely common, and in many cases they make the text unreadable and incomprehensible. Eg p6 “Rather than considering further how cells sort into domains post-differentiation, we will now consider only how cells dynamically update their gene expression as they move between domains of gene expression thereby maintaining coherent domains of gene expression at the tissue level by examining gene expression dynamics during cell movement at the single cell level using a number of examples.”

-Informal, extreme and/or imprecise statements are frequently used. Eg p5 “It would be indeed very surprising if cell movements did not affect Turing patterns at all.” This is a strong statement, explain or remove

-p.5 “Whether or not patterning precision between individual embryos remains robust to differing degrees of cell movement is an important question, particularly as it is hard to imagine how the precise rates and directionality of cell rearrangement can be constrained to a high degree between individuals.”

“the” precise rates – which ones? Why is it hard to imagine?

-Unclear meaning:

--P6 “This potential discrepancy in the mechanism of boundary formation in the presence or absence of cell movements (graded increase in the concentration of single cells versus graded increase in the proportions of expressing cells) might be misleading, making boundaries that form across fields of moving cells appear less precise.”

What is misleading here? Why discrepancy – do you mean difference? It is a matter of the definition of precision – is this what you are trying to say?

--P8 line 37

“Extensive cell mixing early in the culture period was not reported over 2 hours [58], but widespread cell mixing is always observed prior to elongation when observed over 3 hours [57]. “ It is unclear what the authors are trying to say here. Saying “always” is inappropriate. If they want to say that different studies reported conflicting results, then this should be said explicitly. They could discuss what the reasons for this might be.

--p.8 “Further to this feedback from cell movements to patterning, total disaggregation and reassembly of the explant demonstrates that the initial conditions of the cell are not responsible for the final patterning of the explant as would be expected if patterning and cell movements were largely unconnected [57].”

Unclear phrasing – what would be expected? Furthermore, it’s unclear why they call it ‘feedback’ instead of simply an effect.

2. Section 4 reads like a grant proposal rather than a review. The authors present their idea of a pipeline that could help to track and model the role of cell movements in pattern formation. However, they do not review any relevant literature, except for their own preprint (ref 14). In recent years, there have been multiple efforts to measure and model cell interactions and cell

movements in developing tissues, see for instance the examples described here: <https://doi.org/10.1242/dev.151274> . Furthermore, in this section the authors do not provide any systematic discussion of how cell movements can act as a “generative factor”. The content of the section and the title should match.

3. In some instances, phenomena that might be species-specific are discussed in a general way. The relevant species should always be mentioned. Eg
p. 6 line 16 – “In the vertebrate neural tube” – the reference refers to a phenomenon observed in zebrafish. Sorting has not been observed in other species, hence please specify the species.

P 6, paragraph starting on line 47 – please specify which species are being discussed

4. Several statements/descriptions of phenomena are unclear and require more detailed explanation.

-P6, line 24 “Together, these examples demonstrate the process of active cell sorting, where cells are organised into domains, and GRNs can result in high levels of precision and sharp boundaries.” It is unclear how the authors envision that the sorting and the GRN mechanisms work together.

-p.7, top: “Importantly, recent work has demonstrated how the node functions as a stem cell niche to confer stem cell identity to population of resident stem cells [33]. – it is unclear why this is important and how the authors think it is reconciled with the continuous specification of cells that comprise the Node.” The notion of the node being a stem cell niche is to a certain extent contrary to it being continuously reformed – if this is what they want to say, then it should be said explicitly.

-P8 “Despite this extensive cell rearrangement, where every cell’s starting position is randomised, the embryo is still able to undergo a process of self assembly, driven by Nodal signalling [55] acting as a very short range morphogen to reform the embryo at only a single position on the yolk, and initiate germ layer patterning.” It is unclear what exactly happens in this system. Is Nodal signaling acting to re-establish the pattern while the cells are still moving around? If not, then the authors need to be clear about this, because the example would not fall into the category of section 3.

5. Some statements are misleading and need to be corrected:

-p. 8 “The oscillations of Notch signalling, viewed as expression of Her1, have also been demonstrated to be entirely cell autonomous, occurring during in vitro cultures of individual presomitic mesodermal cells [48,49].” The authors say in the next sentences that the oscillations are not entirely cell autonomous – there is communication between the cells. The fact that cells can oscillate in isolation does not mean that the oscillations are autonomous in the normal tissue context.

-P11, line 11 “cell movements themselves have the ability to drive variability in pattern formation 4).” As far as I am aware, this reference doesn’t make this point.

6. Others:

- P8 The authors may want to consider mentioning this paper DOI: [10.1016/j.devcel.2010.06.006](https://doi.org/10.1016/j.devcel.2010.06.006)

-P6, line 16 - this paragraph discusses pattern formation in the presence of movements and hence does not match with the title of this section.

Review form: Reviewer 2

Is the manuscript scientifically sound in its present form?

Yes

Are the interpretations and conclusions justified by the results?

Yes

Is the language acceptable?

Yes

Do you have any ethical concerns with this paper?

No

Have you any concerns about statistical analyses in this paper?

No

Recommendation?

Accept with minor revision (please list in comments)

Comments to the Author(s)

Review of Fulton et al.

The review by Fulton et al. is an interesting discussion of mechanisms of pattern formation during development, taking into account contributions of gene regulatory networks and cell movement. While the review overall is sound, I have a few comments/concerns.

1. Page 3 Lines 38-44: "All of these approaches implicitly depend on being able to assume that gene expression quantified across a tissue over time reflects the gene expression dynamics of the single cells at every position. However, this will only hold true if cells retain (or can at least be assumed to retain) the same coordinates throughout the patterning process. If cells move around and dynamically update their spatial coordinates relative to the tissue over time, tissue-level quantification of gene expression will poorly represent gene expression dynamics in single cells, making it impossible to infer the GRNs driving those dynamics with any reasonable accuracy."

This section is a little confusing to me. I think tissue-level bulk RNA-seq may not be a good representation of single cell gene expression with or without extensive movement, but maybe less so if cells are moving extensively? Single cell-seq has shown that groups of cells originally thought to be relatively homogeneous in their gene expression profiles are often quite heterogeneous.

2. Page 5 Line 6: "the formation of the *tbx16* domain"

tbx16 should be italicized and would be good to indicate that you are referring to mRNA levels here.

3. Page 5 Line 9: "*tbxt16*"

Typo - should be *tbx16*.

4. Page 5 Line 11: "graded increase in the concentration of single cells"

The wording here is unclear as to whether it is an increase in the concentration of tbx16 in single cells, or whether the actual cell density is higher.

5. Page 5 Line 47-48: "One major process during gastrulation (Figure 2A) is the formation of the organiser. The Node is responsible"

A very brief description of what the organiser is would be helpful for readers who are not familiar with this structure, as well as an indication that the organiser is called the node in certain species.

6. Page 5 Line 54-55: "the Node itself in amniotes is a transient structure which is constantly being formed and reformed by the cells that make it up"

The wording here suggests that the node disappears and then reforms, which is not the case, but rather that some cells join and then exit the node.

7. Page 6 Line 16: "as segmental blocks of the presomitic mesoderm are sequentially removed from the anterior, forming somites"

To a reader unfamiliar with somitogenesis, this wording will be a bit confusing – particularly the part about blocks of tissue being removed.

8. Page 6 Figure 2:

The letters (A, B, C) of the figure panels should be referred to in the figure legend.

9. Page 6 Figure 2 legend: "Embryonic Self Assembly in vitro of mouse and human gastruloids"

The figure legend states mouse and human, but the figure shows zebrafish and mouse.

10. Page 7 Line 6: "Through the interaction of Her1/Her7 oscillations"

This is very zebrafish specific. A more general statement could be made about hairy/enhancer of split oscillations.

11. Page 7 Line 10: "the presomitic mesoderm is be achieved"

Typo.

12. Page 7 Line 11: "The oscillations of Notch signalling, viewed as expression of Her1, have also been demonstrated to be entirely cell autonomous, occurring during in vitro cultures of individual presomitic mesodermal cells"

The oscillations of Her1 occur in the absence of Notch signaling (see Delaune et al 2012 Dev Cell and others), but are asynchronous. Notch signaling synchronizes the oscillations between cells (but her1 oscillation alone is not a readout of Notch signaling oscillations).

13. Reference 54

The title of this reference is missing.

14. Page 7 Line 31-35: "This process of self assembly, where cells reaggregate and establish a pattern despite enormous prior mixing is not unique to the annual killifish and has been

demonstrated through whole embryonic explants of Atlantic Killifish [56], zebrafish [57,58] and cavefish [59] embryos.”

The wording here implies that killifish, zebrafish, and cavefish normally undergo self-assembly, but it should be made clear that they have the capacity to do so when forced to experimentally.

15. Page 8 Line 21-25: “As described previously, these approaches have been very successful and demonstrate the potential power of morphogen gradients in the patterning of tissues however cannot be used to understand pattern formation in the presence of extensive cell mixing.”

In some cases these presumably can still be used as long as the GRN of mixing cells can rapidly respond to changes in the morphogen gradient as they move to a different position.

16. Page 8 Line 35-36: “or move posteriorly, away from the differentiation front in the zebrafish presomitic mesoderm.”

It is not clear which are the cells moving away from the differentiation front, is there a reference for this?

17. Page 9 Line 40-44: “The resultant signalling response to perturbation can also be measured and quantified. With these results in addition to tracks and signalling profiles from untreated embryos, a number of simulations can be run to disentangle the role of signalling in morphogenesis from cell fate decision making. By comparison of the results of simulations using signalling perturbed tracks versus wild type tracks with either wild type signals or perturbed signalling as inputs to a single conserved GRN, it may be possible to uncouple these two functions of a single signal such as FGF.”

The wording here makes it a bit hard to understand what exactly is being proposed and how it would be enacted experimentally. What is meant by “signalling response to perturbation” here and “signalling profiles from untreated embryos?”

18. Figure 3:

The suggestion here is to develop a gene regulatory network and then map the network onto cell tracking data. Something that is described in the text but missing in the figure is the need to map the patterns of signalling across the tissue. The panel on the bottom right also shows the GRN being rapidly modified as the cell moves through the tissue. However, this depiction seems more consistent with the traditional morphogen model, where despite the movement the cell can rapidly change its gene expression profile based on the new signaling/morphogen concentration environment.

19. Page 10 Line 6-7: “difference are likely to be, in part, due to changes in the morphogenesis and embryonic geometry of one species compared to another, and not only due to the strength of interactions between network nodes.”

At some level one would expect the signalling and GRN controlling fate to also impact morphogenesis, so if the signalling and GRN are held constant between species during the modelling how exactly do the differences arise? Is the suggestion here that cells in different species have inherently different migratory/behavioral properties, either based on cell size or starting cell density, and/or that the space in which the cells are able to travel differs in such a way that it impacts the migratory dynamics of the population? Whatever the hypothesis may be, I think it could be made more explicit here.

Decision letter (RSOS-211293.R0)

Dear Dr Steventon

The Editors assigned to your paper RSOS-211293 "The Unappreciated Generative Role of Cell Movements in Pattern" have now received comments from reviewers and would like you to revise the paper in accordance with the reviewer comments and any comments from the Editors. Please note this decision does not guarantee eventual acceptance.

Please submit your revised manuscript and required files (see below) no later than Friday 14 January 2022. Note: the ScholarOne system will 'lock' if submission of the revision is attempted after the deadline. If you do not think you will be able to meet this deadline please contact the editorial office immediately.

on behalf of Professor Andrew Copp (Associate Editor) and Malcolm White (Subject Editor)
openscience@royalsociety.org

Associate Editor Comments to Author (Professor Andrew Copp):

Associate Editor: 1

Comments to the Author:

Your paper has now been seen by two reviewers, both of whom consider the topic to be novel, interesting and important within the field of developmental biology. However, as you will see, both reviewers have a number of specific criticisms of the manuscript in its present form. If you are able to address these criticisms in a revised manuscript, I am happy to re-consider the paper publication.

Reviewer comments to Author:

Reviewer: 1

Comments to the Author(s)

In their manuscript, Fulton et al review an interesting current question in developmental biology - what is the role of cell movements in pattern formation. Recent advances in imaging and

mathematical modeling have allowed new insights to be gained into this question, hence the review is timely and relevant. The idea and general structure of the manuscript are well conceived. However, as I outline below, there are major issues that make the manuscript unsuitable for publication in its present form.

1. The writing quality is poor in places. Grammatical errors often make it impossible to understand what the authors are trying to say. I am giving several examples below, but the text will need to be thoroughly edited.

-Run on sentences are extremely common, and in many cases they make the text unreadable and incomprehensible. Eg p6 "Rather than considering further how cells sort into domains post-differentiation, we will now consider only how cells dynamically update their gene expression as they move between domains of gene expression thereby maintaining coherent domains of gene expression at the tissue level by examining gene expression dynamics during cell movement at the single cell level using a number of examples."

-Informal, extreme and/or imprecise statements are frequently used. Eg p5 "It would be indeed very surprising if cell movements did not affect Turing patterns at all." This is a strong statement, explain or remove

-p.5 "Whether or not patterning precision between individual embryos remains robust to differing degrees of cell movement is an important question, particularly as it is hard to imagine how the precise rates and directionality of cell rearrangement can be constrained to a high degree between individuals."

"the" precise rates – which ones? Why is it hard to imagine?

-Unclear meaning:

--P6 "This potential discrepancy in the mechanism of boundary formation in the presence or absence of cell movements (graded increase in the concentration of single cells versus graded increase in the proportions of expressing cells) might be misleading, making boundaries that form across fields of moving cells appear less precise."

What is misleading here? Why discrepancy – do you mean difference? It is a matter of the definition of precision – is this what you are trying to say?

--P8 line 37

"Extensive cell mixing early in the culture period was not reported over 2 hours [58], but widespread cell mixing is always observed prior to elongation when observed over 3 hours [57]. " It is unclear what the authors are trying to say here. Saying "always" is inappropriate. If they want to say that different studies reported conflicting results, then this should be said explicitly. They could discuss what the reasons for this might be.

--p.8 "Further to this feedback from cell movements to patterning, total disaggregation and reassembly of the explant demonstrates that the initial conditions of the cell are not responsible for the final patterning of the explant as would be expected if patterning and cell movements were largely unconnected [57]."

Unclear phrasing – what would be expected? Furthermore, it's unclear why they call it 'feedback' instead of simply an effect.

2. Section 4 reads like a grant proposal rather than a review. The authors present their idea of a pipeline that could help to track and model the role of cell movements in pattern formation. However, they do not review any relevant literature, except for their own preprint (ref 14). In recent years, there have been multiple efforts to measure and model cell interactions and cell movements in developing tissues, see for instance the examples described here: <https://doi.org/10.1242/dev.151274> . Furthermore, in this section the authors do not provide any systematic discussion of how cell movements can act as a "generative factor". The content of the section and the title should match.

3. In some instances, phenomena that might be species-specific are discussed in a general way. The relevant species should always be mentioned. Eg
 p. 6 line 16 - "In the vertebrate neural tube" - the reference refers to a phenomenon observed in zebrafish. Sorting has not been observed in other species, hence please specify the species.

P 6, paragraph starting on line 47 - please specify which species are being discussed

4. Several statements/descriptions of phenomena are unclear and require more detailed explanation.

-P6, line 24 "Together, these examples demonstrate the process of active cell sorting, where cells are organised into domains, and GRNs can result in high levels of precision and sharp boundaries." It is unclear how the authors envision that the sorting and the GRN mechanisms work together.

-p.7, top: "Importantly, recent work has demonstrated how the node functions as a stem cell niche to confer stem cell identity to population of resident stem cells [33]. - it is unclear why this is important and how the authors think it is reconciled with the continuous specification of cells that comprise the Node." The notion of the node being a stem cell niche is to a certain extent contrary to it being continuously reformed - if this is what they want to say, then it should be said explicitly.

-P8 "Despite this extensive cell rearrangement, where every cell's starting position is randomised, the embryo is still able to undergo a process of self assembly, driven by Nodal signalling [55] acting as a very short range morphogen to reform the embryo at only a single position on the yolk, and initiate germ layer patterning." It is unclear what exactly happens in this system. Is Nodal signaling acting to re-establish the pattern while the cells are still moving around? If not, then the authors need to be clear about this, because the example would not fall into the category of section 3.

5. Some statements are misleading and need to be corrected:

-p. 8 "The oscillations of Notch signalling, viewed as expression of Her1, have also been demonstrated to be entirely cell autonomous, occurring during in vitro cultures of individual presomitic mesodermal cells [48,49]." The authors say in the next sentences that the oscillations are not entirely cell autonomous - there is communication between the cells. The fact that cells can oscillate in isolation does not mean that the oscillations are autonomous in the normal tissue context.

-P11, line 11 "cell movements themselves have the ability to drive variability in pattern formation 4)." As far as I am aware, this reference doesn't make this point.

6. Others:

- P8 The authors may want to consider mentioning this paper DOI: 10.1016/j.devcel.2010.06.006

-P6, line 16 - this paragraph discusses pattern formation in the presence of movements and hence does not match with the title of this section.

Reviewer: 2

Comments to the Author(s)

Review of Fulton et al.

The review by Fulton et al. is an interesting discussion of mechanisms of pattern formation during development, taking into account contributions of gene regulatory networks and cell movement. While the review overall is sound, I have a few comments/concerns.

1. Page 3 Lines 38-44: “All of these approaches implicitly depend on being able to assume that gene expression quantified across a tissue over time reflects the gene expression dynamics of the single cells at every position. However, this will only hold true if cells retain (or can at least be assumed to retain) the same coordinates throughout the patterning process. If cells move around and dynamically update their spatial coordinates relative to the tissue over time, tissue-level quantification of gene expression will poorly represent gene expression dynamics in single cells, making it impossible to infer the GRNs driving those dynamics with any reasonable accuracy.”

This section is a little confusing to me. I think tissue-level bulk RNA-seq may not be a good representation of single cell gene expression with or without extensive movement, but maybe less so if cells are moving extensively? Single cell-seq has shown that groups of cells originally thought to be relatively homogeneous in their gene expression profiles are often quite heterogeneous.

2. Page 5 Line 6: “the formation of the *tbx16* domain”

tbx16 should be italicized and would be good to indicate that you are referring to mRNA levels here.

3. Page 5 Line 9: “*tbxt16*”

Typo - should be *tbx16*.

4. Page 5 Line 11: “graded increase in the concentration of single cells”

The wording here is unclear as to whether it is an increase in the concentration of *tbx16* in single cells, or whether the actual cell density is higher.

5. Page 5 Line 47-48: “One major process during gastrulation (Figure 2A) is the formation of the organiser. The Node is responsible”

A very brief description of what the organiser is would be helpful for readers who are not familiar with this structure, as well as an indication that the organiser is called the node in certain species.

6. Page 5 Line 54-55: “the Node itself in amniotes is a transient structure which is constantly being formed and reformed by the cells that make it up”

The wording here suggests that the node disappears and then reforms, which is not the case, but rather that some cells join and then exit the node.

7. Page 6 Line 16: “as segmental blocks of the presomitic mesoderm are sequentially removed from the anterior, forming somites”

To a reader unfamiliar with somitogenesis, this wording will be a bit confusing - particularly the part about blocks of tissue being removed.

8. Page 6 Figure 2:

The letters (A, B, C) of the figure panels should be referred to in the figure legend.

9. Page 6 Figure 2 legend: “Embryonic Self Assembly in vitro of mouse and human gastruloids”

The figure legend states mouse and human, but the figure shows zebrafish and mouse.

10. Page 7 Line 6: "Through the interaction of Her1/Her7 oscillations"

This is very zebrafish specific. A more general statement could be made about hairy/enhancer of split oscillations.

11. Page 7 Line 10: "the presomitic mesoderm is be achieved"

Typo.

12. Page 7 Line 11: : "The oscillations of Notch signalling, viewed as expression of Her1, have also been demonstrated to be entirely cell autonomous, occurring during in vitro cultures of individual presomitic mesodermal cells"

The oscillations of Her1 occur in the absence of Notch signaling (see Delaune et al 2012 Dev Cell and others), but are asynchronous. Notch signaling synchronizes the oscillations between cells (but her1 oscillation alone is not a readout of Notch signaling oscillations).

13. Reference 54

The title of this reference is missing.

14. Page 7 Line 31-35: "This process of self assembly, where cells reaggregate and establish a pattern despite enormous prior mixing is not unique to the annual killifish and has been demonstrated through whole embryonic explants of Atlantic Killifish [56], zebrafish [57,58] and cavefish [59] embryos."

The wording here implies that killifish, zebrafish, and cavefish normally undergo self-assembly, but it should be made clear that they have the capacity to do so when forced to experimentally.

15. Page 8 Line 21-25: "As described previously, these approaches have been very successful and demonstrate the potential power of morphogen gradients in the patterning of tissues however cannot be used to understand pattern formation in the presence of extensive cell mixing."

In some cases these presumably can still be used as long as the GRN of mixing cells can rapidly respond to changes in the morphogen gradient as they move to a different position.

16. Page 8 Line 35-36: "or move posteriorly, away from the differentiation front in the zebrafish presomitic mesoderm."

It is not clear which are the cells moving away from the differentiation front, is there a reference for this?

17. Page 9 Line 40-44: "The resultant signalling response to perturbation can also be measured and quantified. With these results in addition to tracks and signalling profiles from untreated embryos, a number of simulations can be run to disentangle the role of signalling in morphogenesis from cell fate decision making. By comparison of the results of simulations using signalling perturbed tracks versus wild type tracks with either wild type signals or perturbed signalling as inputs to a single conserved GRN, it may be possible to uncouple these two functions of a single signal such as FGF."

The wording here makes it a bit hard to understand what exactly is being proposed and how it would be enacted experimentally. What is meant by “signalling response to perturbation” here and “signalling profiles from untreated embryos?”

18. Figure 3:

The suggestion here is to develop a gene regulatory network and then map the network onto cell tracking data. Something that is described in the text but missing in the figure is the need to map the patterns of signalling across the tissue. The panel on the bottom right also shows the GRN being rapidly modified as the cell moves through the tissue. However, this depiction seems more consistent with the traditional morphogen model, where despite the movement the cell can rapidly change its gene expression profile based on the new signaling/morphogen concentration environment.

19. Page 10 Line 6-7: “difference are likely to be, in part, due to changes in the morphogenesis and embryonic geometry of one species compared to another, and not only due to the strength of interactions between network nodes.”

At some level one would expect the signalling and GRN controlling fate to also impact morphogenesis, so if the signalling and GRN are held constant between species during the modelling how exactly do the differences arise? Is the suggestion here that cells in different species have inherently different migratory/behavioral properties, either based on cell size or starting cell density, and/or that the space in which the cells are able to travel differs in such a way that it impacts the migratory dynamics of the population? Whatever the hypothesis may be, I think it could be made more explicit here.

===PREPARING YOUR MANUSCRIPT===

If you have been asked to revise the written English in your submission as a condition of publication, you must do so, and you are expected to provide evidence that you have received language editing support. The journal would prefer that you use a professional language editing service and provide a certificate of editing, but a signed letter from a colleague who is a fluent

speaker of English is acceptable. Note the journal has arranged a number of discounts for authors using professional language editing services (<https://royalsociety.org/journals/authors/benefits/language-editing/>).

===PREPARING YOUR REVISION IN SCHOLARONE===

<https://royalsociety.org/journals/authors/author-guidelines/#supplementary-material> to include a suitable title and informative caption. An example of appropriate titling and captioning may be found at https://figshare.com/articles/Table_S2_from_Is_there_a_trade-

off_between_peak_performance_and_performance_breadth_across_temperatures_for_aerobic_sc
ope_in_teleost_fishes_/3843624.

Author's Response to Decision Letter for (RSOS-211293.R0)

See Appendix A.

RSOS-211293.R1 (Revision)

Review form: Reviewer 1

Is the manuscript scientifically sound in its present form?

Yes

Are the interpretations and conclusions justified by the results?

Yes

Is the language acceptable?

Yes

Do you have any ethical concerns with this paper?

No

Have you any concerns about statistical analyses in this paper?

No

Recommendation?

Accept with minor revision (please list in comments)

Comments to the Author(s)

The authors have improved the manuscript. I have the following minor comments:

- 1) "Minimal cell rearrangement is not observed over the first 2 hours post culture [57], but appears to occur later as the explants become polarised with respect to germ layer markers [56]. " This is not understandable. What is 'minimal', why minimal?
- 2) "Furthermore, multi-dimensional modelling of plant development has revealed significant insights into the intersections of tissue morphogenesis and patterning []." Missing reference

Review form: Reviewer 2

Is the manuscript scientifically sound in its present form?

Yes

Are the interpretations and conclusions justified by the results?

Yes

Is the language acceptable?

Yes

Do you have any ethical concerns with this paper?

No

Have you any concerns about statistical analyses in this paper?

No

Recommendation?

Accept as is

Comments to the Author(s)

The authors have adequately addressed my concerns and I now support publication.

Decision letter (RSOS-211293.R1)

Dear Dr Steventon

On behalf of the Editors, we are pleased to inform you that your Manuscript RSOS-211293.R1 "The Unappreciated Generative Role of Cell Movements in Pattern" has been accepted for publication in Royal Society Open Science subject to minor revision in accordance with the referees' reports. Please find the referees' comments along with any feedback from the Editors below my signature.

Please submit your revised manuscript and required files (see below) no later than 7 days from today's (ie 29-Mar-2022) date. Note: the ScholarOne system will 'lock' if submission of the revision is attempted 7 or more days after the deadline. If you do not think you will be able to meet this deadline please contact the editorial office immediately.

Please note article processing charges apply to papers accepted for publication in Royal Society Open Science (<https://royalsocietypublishing.org/rsos/charges>). Charges will also apply to papers transferred to the journal from other Royal Society Publishing journals, as well as papers

submitted as part of our collaboration with the Royal Society of Chemistry (<https://royalsocietypublishing.org/rsos/chemistry>). Fee waivers are available but must be requested when you submit your revision (<https://royalsocietypublishing.org/rsos/waivers>).

on behalf of Professor Andrew Copp (Associate Editor) and Malcolm White (Subject Editor)
openscience@royalsociety.org

Reviewer comments to Author

Reviewer: 1

Comments to the Author(s)

The authors have improved the manuscript. I have the following minor comments:

- 1) "Minimal cell rearrangement is not observed over the first 2 hours post culture [57], but appears to occur later as the explants become polarised with respect to germ layer markers [56]. " This is not understandable. What is 'minimal', why minimal?
- 2) "Furthermore, multi-dimensional modelling of plant development has revealed significant insights into the intersections of tissue morphogenesis and patterning []." Missing reference

Reviewer: 2

Comments to the Author(s)

The authors have adequately addressed my concerns and I now support publication.

===PREPARING YOUR MANUSCRIPT===

- one version should clearly identify all the changes that have been made (for instance, in coloured highlight, in bold text, or tracked changes);
- a 'clean' version of the new manuscript that incorporates the changes made, but does not highlight them. This version will be used for typesetting.

While not essential, it will speed up the preparation of your manuscript proof if you format your references/bibliography in Vancouver style (please see

<https://royalsociety.org/journals/authors/author-guidelines/#formatting>). You should include DOIs for as many of the references as possible.

===PREPARING YOUR REVISION IN SCHOLARONE===

<https://royalsociety.org/journals/authors/author-guidelines/#data>. You should ensure that you cite the dataset in your reference list. If you have deposited data etc in the Dryad repository,

please only include the 'For publication' link at this stage. You should remove the 'For review' link.

-- If you are requesting an article processing charge waiver, you must select the relevant waiver option (if requesting a discretionary waiver, the form should have been uploaded, see 'File upload' above).

-- If you have uploaded any electronic supplementary (ESM) files, please ensure you follow the guidance at <https://royalsociety.org/journals/authors/author-guidelines/#supplementary-material> to include a suitable title and informative caption. An example of appropriate titling and captioning may be found at https://figshare.com/articles/Table_S2_from_Is_there_a_trade-off_between_peak_performance_and_performance_breadth_across_temperatures_for_aerobic_scope_in_teleost_fishes_/3843624.

Author's Response to Decision Letter for (RSOS-211293.R1)

See Appendix B.

Decision letter (RSOS-211293.R2)

Dear Dr Steventon,

I am pleased to inform you that your manuscript entitled "The Unappreciated Generative Role of Cell Movements in Pattern" is now accepted for publication in Royal Society Open Science.

Please see the Royal Society Publishing guidance on how you may share your accepted author manuscript at <https://royalsociety.org/journals/ethics-policies/media-embargo/>. After publication, some additional ways to effectively promote your article can also be found here

<https://royalsociety.org/blog/2020/07/promoting-your-latest-paper-and-tracking-your-results/>.

on behalf of Professor Andrew Copp (Associate Editor) and Professor Malcolm White (Subject Editor)

Appendix A

Reviewer comments to Author:

Reviewer: 1

Comments to the Author(s)

In their manuscript, Fulton et al review an interesting current question in developmental biology – what is the role of cell movements in pattern formation. Recent advances in imaging and mathematical modeling have allowed new insights to be gained into this question, hence the review is timely and relevant. The idea and general structure of the manuscript are well conceived. However, as I outline below, there are major issues that make the manuscript unsuitable for publication in its present form.

1. The writing quality is poor in places. Grammatical errors often make it impossible to understand what the authors are trying to say. I am giving several examples below, but the text will need to be thoroughly edited.

Answer:

We have been through the article and corrected multiple grammatical errors in line with this reviewer's suggestion.

-Run on sentences are extremely common, and in many cases they make the text unreadable and incomprehensible. Eg p6 "Rather than considering further how cells sort into domains post-differentiation, we will now consider only how cells dynamically update their gene expression as they move between domains of gene expression thereby maintaining coherent domains of gene expression at the tissue level by examining gene expression dynamics during cell movement at the single cell level using a number of examples."

Answer:

This sentence has been clarified, and we have gone through the text to adjust any additional run-on sentences.

-Informal, extreme and/or imprecise statements are frequently used. Eg p5 "It would be indeed very surprising if cell movements did not affect Turing patterns at all." This is a strong statement, explain or remove

Answer:

This statement has been removed

-p.5 "Whether or not patterning precision between individual embryos remains robust to differing degrees of cell movement is an important question, particularly as it is hard to imagine how the precise rates and directionality of cell rearrangement

can be constrained to a high degree between individuals.”
“the” precise rates – which ones? Why is it hard to imagine?

Answer:

We have now expanded this point to a paragraph to ensure the full meaning of the sentence is fully communicated.

-Unclear meaning:

--P6 “This potential discrepancy in the mechanism of boundary formation in the presence or absence of cell movements (graded increase in the concentration of single cells versus graded increase in the proportions of expressing cells) might be misleading, making boundaries that form across fields of moving cells appear less precise.”

What is misleading here? Why discrepancy – do you mean difference? It is a matter of the definition of precision – is this what you are trying to say?

Answer:

This paragraph has now been removed in response the additional valid point of the reviewer made above.

--P8 line 37

“Extensive cell mixing early in the culture period was not reported over 2 hours [58], but widespread cell mixing is always observed prior to elongation when observed over 3 hours [57].”

It is unclear what the authors are trying to say here. Saying “always” is inappropriate. If they want to say that different studies reported conflicting results, then this should be said explicitly. They could discuss what the reasons for this might be.

Answer:

We have now clarified this statement to make it clear that the studies are dealing with different time-periods post culture.

--p.8 “Further to this feedback from cell movements to patterning, total disaggregation and reassembly of the explant demonstrates that the initial conditions of the cell are not responsible for the final patterning of the explant as would be expected if patterning and cell movements were largely unconnected [57].”
Unclear phrasing – what would be expected? Furthermore, it’s unclear why they call it ‘feedback’ instead of simply an effect.

Answer:

This section has been re-written to highlight a lack of knowledge surrounding the self-organising potential of germ layer patterning in early zebrafish development.

2. Section 4 reads like a grant proposal rather than a review. The authors present their idea of a pipeline that could help to track and model the role of cell movements in pattern formation. However, they do not review any relevant literature, except for their own preprint (ref 14). In recent years, there have been multiple efforts to measure and model cell interactions and cell movements in developing tissues, see for instance the examples described here: <https://doi.org/10.1242/dev.151274> .

Answer:

We have inserted an additional paragraph that refers to specific cases where tissue morphogenesis and pattern formation have been addressed in the context of multi-scalar computational models, and referred to the suggested review for further reading. We highlight how the specific instances dealt with in the context of this review go beyond the capabilities of such approaches due to the number of cells involved. Finally, we have re-worded sections of this article to refer to cited research articles only, rather than to 'propose' one particular strategy over another.

Furthermore, in this section the authors do not provide any systematic discussion of how cell movements can act as a "generative factor". The content of the section and the title should match.

Answer:

We have changed the title to more accurately reflect the contents of this section.

3. In some instances, phenomena that might be species-specific are discussed in a general way. The relevant species should always be mentioned. Eg p. 6 line 16 – "In the vertebrate neural tube" – the reference refers to a phenomenon observed in zebrafish. Sorting has not been observed in other species, hence please specify the species.

Answer:

This has now been corrected

P 6, paragraph starting on line 47 – please specify which species are being discussed

Answer:

This has been clarified

4. Several statements/descriptions of phenomena are unclear and require more detailed explanation.

-P6, line 24 "Together, these examples demonstrate the process of active cell sorting, where cells are organised into domains, and GRNs can result in high levels of precision and sharp boundaries. " It is unclear how the authors envision that the

sorting and the GRN mechanisms work together.

Answer:

This has been clarified- we simply mean that distinct gene expression profiles lead to differential cell adhesion molecule expression, and subsequently cell sorting.

-p.7, top: "Importantly, recent work has demonstrated how the node functions as a stem cell niche to confer stem cell identity to population of resident stem cells [33]. – it is unclear why this is important and how the authors think it is reconciled with the continuous specification of cells that comprise the Node." The notion of the node being a stem cell niche is to a certain extent contrary to it being continuously reformed – if this is what they want to say, then it should be said explicitly.

Answer:

The cited work demonstrates how cells become exposed to a stem cell niche as they move into the node, and provides an explanation for how cells adopt an organiser state. This has now been clarified

-P8 "Despite this extensive cell rearrangement, where every cell's starting position is randomised, the embryo is still able to undergo a process of self assembly, driven by Nodal signalling [55] acting as a very short range morphogen to reform the embryo at only a single position on the yolk, and initiate germ layer patterning." It is unclear what exactly happens in this system. Is Nodal signaling acting to re-establish the pattern while the cells are still moving around? If not, then the authors need to be clear about this, because the example would not fall into the category of section 3.

Answer:

This section aims to give examples of how gene expression patterns can emerge even when cells are continually rearranging in a manner that would otherwise be expected to disrupt pattern formation. The precise role of Nodal signalling in the above-mentioned context is not yet known, and we are highlighting the question here as an important area for future research. We have added a statement at the end of the following revised paragraph that clarifies the need to better understand the role of Nodal in such contexts.

5. Some statements are misleading and need to be corrected:

-p. 8 "The oscillations of Notch signalling, viewed as expression of Her1, have also been demonstrated to be entirely cell autonomous, occurring during in vitro cultures of individual presomitic mesodermal cells [48,49]." The authors say in the next sentences that the oscillations are not entirely cell autonomous – there is communication between the cells. The fact that cells can oscillate in isolation does not mean that the oscillations are autonomous in the normal tissue context.

Answer:

We have re-phrased this to make it clear that the experiments cited demonstrate a cell autonomous oscillation *in vitro*. We go onto discuss how non-autonomous signals are required to synchronise oscillations such that waves are generated. Therefore, the overall patterning mechanism is a composite of both autonomous and non-autonomous processes, with cell movements proposed to play an important role in ensuring that cells contact one another for the appropriate length of time.

-P11, line 11 "cell movements themselves have the ability to drive variability in pattern formation 4). " As far as I am aware, this reference doesn't make this point.

Answer:

We have clarified this to make it clear that we are proposing future directions that could be taken though the line of research discussed.

6. Others:

- P8 The authors may want to consider mentioning this paper DOI: 10.1016/j.devcel.2010.06.006

Answer:

We believe that introducing a full discussion of lateral inhibition mechanisms in development is beyond the scope of this article

-P6, line 16 - this paragraph discusses pattern formation in the presence of movements and hence does not match with the title of this section.

Answer:

We are contrasting systems in which local cell sorting can refine a pattern, versus those where extensive cell movements must be accommodated for if gene expression patterns are to be maintained.

Reviewer: 2

Comments to the Author(s)

Review of Fulton et al.

The review by Fulton et al. is an interesting discussion of mechanisms of pattern formation during development, taking into account contributions of gene regulatory networks and cell movement. While the review overall is sound, I have a few comments/concerns.

1. Page 3 Lines 38-44: "All of these approaches implicitly depend on being able to assume that gene expression quantified across a tissue over time reflects the gene expression dynamics of the single cells at every position. However, this will only hold true if cells retain (or can at least be assumed to retain) the same coordinates throughout the patterning process. If cells move around and dynamically update their spatial coordinates relative to the tissue over time, tissue-level quantification of gene expression will poorly represent gene expression dynamics in single cells, making it impossible to infer the GRNs driving those dynamics with any reasonable accuracy."

This section is a little confusing to me. I think tissue-level bulk RNA-seq may not be a good representation of single cell gene expression with or without extensive movement, but maybe less so if cells are moving extensively? Single cell-seq has shown that groups of cells originally thought to be relatively homogeneous in their gene expression profiles are often quite heterogeneous.

Answer:

Thank you for pointing this out. We realise now that there are many ways to quantify gene expression and that therefore what we meant was not clear. The kind of quantification that we are referring to here is the quantification of spatio-temporal gene expression patterns across a tissue using staining techniques. To clarify this, the first sentence in the paragraph has been changed to: "*All of these approaches implicitly depend on being able to assume that the quantification of spatio-temporal gene expression patterns across a tissue reflects the gene expression dynamics of the single cells at every position.*"

2. Page 5 Line 6: "the formation of the *tbx16* domain"

tbx16 should be italicized and would be good to indicate that you are referring to mRNA levels here.

Answer

This has now been corrected everywhere

3. Page 5 Line 9: "tbxt16"

Typo – should be tbx16.

Answer

This has now been corrected everywhere

4. Page 5 Line 11: "graded increase in the concentration of single cells"

The wording here is unclear as to whether it is an increase in the concentration of tbx16 in single cells, or whether the actual cell density is higher.

Answer

This has now been removed as part of edits in response to another reviewer's comment.

5. Page 5 Line 47-48: "One major process during gastrulation (Figure 2A) is the formation of the organiser. The Node is responsible"

A very brief description of what the organiser is would be helpful for readers who are not familiar with this structure, as well as an indication that the organiser is called the node in certain species.

Answer

We have now changed these sentences to include a brief definition of the organiser and to clarify that the node is what we call the organiser in avian models. It now reads: "One major process during gastrulation (Figure \ref{fig:Cell Mixing}A) is the formation of the organiser; a group of cells within the embryo that are able to determine the fate and morphogenesis of cells around them (Martinez Arias et al 2018). The Node (the organizer in avian species)...."

6. Page 5 Line 54-55: "the Node itself in amniotes is a transient structure which is constantly being formed and reformed by the cells that make it up"

The wording here suggests that the node disappears and then reforms, which is not the case, but rather that some cells join and then exit the node.

Answer

This sentence has now been changed to: "This therefore suggests that while the Node itself persists during development, its components are transient, with cells

constantly entering and exiting it.”

7. Page 6 Line 16: “as segmental blocks of the presomitic mesoderm are sequentially removed from the anterior, forming somites”

To a reader unfamiliar with somitogenesis, this wording will be a bit confusing – particularly the part about blocks of tissue being removed.

Answer

This has now been changed to: “(...)as the somites (segmented blocks of mesoderm) bud off sequentially from the anterior presomitic mesoderm..”

8. Page 6 Figure 2:

The letters (A, B, C) of the figure panels should be referred to in the figure legend.

Answer

This has now been corrected

9. Page 6 Figure 2 legend: “Embryonic Self Assembly in vitro of mouse and human gastruloids”

The figure legend states mouse and human, but the figure shows zebrafish and mouse.

Answer

This has now been changed to “embryonic self assembly in vitro of zebrafish pescoids, made from whole embryonic explants taken prior to gastrulation, and mouse gastruloids, made from embryonic stem cells. Human gastruloids are made in the same way as mouse gastruloids”

10. Page 7 Line 6: “Through the interaction of Her1/Her7 oscillations”

This is very zebrafish specific. A more general statement could be made about hairy/enhancer of split oscillations.

Answer

This has now been changed to reflect the evolutionary plasticity of the segmentation clock to: “Through the interaction of segmentation clock gene oscillations (Her1/Her7 oscillations in zebrafish)..”

11. Page 7 Line 10: “the presomitic mesoderm is be achieved”

Typo.

Answer

Corrected, thanks.

12. Page 7 Line 11: : "The oscillations of Notch signalling, viewed as expression of Her1, have also been demonstrated to be entirely cell autonomous, occurring during in vitro cultures of individual presomitic mesodermal cells"

The oscillations of Her1 occur in the absence of Notch signaling (see Delaune et al 2012 Dev Cell and others), but are asynchronous. Notch signaling synchronizes the oscillations between cells (but her1 oscillation alone is not a readout of Notch signaling oscillations).

Answer

This sentence has now been changed to " The oscillations of Her 1, which in the embryo are synchronised by Notch signalling, have also been demonstrated to be entirely cell autonomous, occurring during *in vitro* cultures of individual presomitic mesodermal cells \cite{Webb2016, Rohde2021Cell-autonomousClock}."

13. Reference 54

The title of this reference is missing.

Answer

This has been corrected.

14. Page 7 Line 31-35: "This process of self assembly, where cells reaggregate and establish a pattern despite enormous prior mixing is not unique to the annual killifish and has been demonstrated through whole embryonic explants of Atlantic Killifish [56], zebrafish [57,58] and cavefish [59] embryos."

The wording here implies that killifish, zebrafish, and cavefish normally undergo self-assembly, but it should be made clear that they have the capacity to do so when forced to experimentally.

Answer

This has now been reworded to " This process of self assembly, where cells reaggregate and establish a pattern is not unique to the annual killifish and has also been observed experimentally after dissociating whole embryonic explants of Atlantic Killifish \cite{Oppenheimer1936a}, zebrafish \cite{Fulton2020, Schauer2020} and cavefish \cite{Torres-Paz2020} embryos" to highlight the experimental

manipulation

15. Page 8 Line 21-25: "As described previously, these approaches have been very successful and demonstrate the potential power of morphogen gradients in the patterning of tissues however cannot be used to understand pattern formation in the presence of extensive cell mixing."

In some cases these presumably can still be used as long as the GRN of mixing cells can rapidly respond to changes in the morphogen gradient as they move to a different position.

Answer

We have now added "in general" to highlight that when cells are mixing, in general, morphogen gradients will not work like we expect them to. We have however not added a mention here to special cases like the one the reviewer suggested as we don't want our reader to focus on special cases, which have been referred to elsewhere in the text. As such, this sentence now reads: "As described previously, these approaches have been very successful and demonstrate the potential power of morphogen gradients in the patterning of tissues however cannot be used in general to understand pattern formation in the presence of extensive cell mixing."

16. Page 8 Line 35-36: "or move posteriorly, away from the differentiation front in the zebrafish presomitic mesoderm."

It is not clear which are the cells moving away from the differentiation front, is there a reference for this?

Answer

This is our own work, which has now been referenced accordingly

17. Page 9 Line 40-44: "The resultant signalling response to perturbation can also be measured and quantified. With these results in addition to tracks and signalling profiles from untreated embryos, a number of simulations can be run to disentangle the role of signalling in morphogenesis from cell fate decision making. By comparison of the results of simulations using signalling perturbed tracks versus wild type tracks with either wild type signals or perturbed signalling as inputs to a single conserved GRN, it may be possible to uncouple these two functions of a single signal such as FGF."

The wording here makes it a bit hard to understand what exactly is being proposed and how it would be enacted experimentally. What is meant by "signalling response to perturbation" here and "signalling profiles from untreated embryos?"

Answer

This section has now been rewritten:

“The resulting signalling profiles in these embryos can then be quantified and used to simulate pattern formation in a perturbed signalling scenario which takes into account the effect of signalling on cell movements. These simulations can then be directly compared to those obtained using wild type signalling profiles and tracks, to help disentangle the role of signalling in morphogenesis and cell fate decision making.”

18. Figure 3:

The suggestion here is to develop a gene regulatory network and then map the network onto cell tracking data. Something that is described in the text but missing in the figure is the need to map the patterns of signalling across the tissue. The panel on the bottom right also shows the GRN being rapidly modified as the cell moves through the tissue. However, this depiction seems more consistent with the traditional morphogen model, where despite the movement the cell can rapidly change its gene expression profile based on the new signaling/morphogen concentration environment.

Answer

We have now added some shading to the tailbuds in Fig 3 to illustrate that we project the quantified signals onto the tracks.

The GRN is not being modified *per se*, however the fate of the cell does change as it moves towards a somite. What we are trying to highlight here is that this is not a direct read out of the morphogen gradient, but rather, it is mediated by the cell's movements.

19. Page 10 Line 6-7: “difference are likely to be, in part, due to changes in the morphogenesis and embryonic geometry of one species compared to another, and not only due to the strength of interactions between network nodes.”

At some level one would expect the signalling and GRN controlling fate to also impact morphogenesis, so if the signalling and GRN are held constant between species during the modelling how exactly do the differences arise? Is the suggestion here that cells in different species have inherently different migratory/behavioral properties, either based on cell size or starting cell density, and/or that the space in which the cells are able to travel differs in such a way that it impacts the migratory dynamics of the population? Whatever the hypothesis may be, I think it could be made more explicit here.

Answer

These are exactly the sort of additional factors we mean in the context of the discussion, and are likely not under the control of GRN interactions that are the principle focus of patterning studies. We have increased the clarity of this statement.

===PREPARING YOUR MANUSCRIPT===

- one version identifying all the changes that have been made (for instance, in coloured highlight, in bold text, or tracked changes);
- a 'clean' version of the new manuscript that incorporates the changes made, but does not highlight them. This version will be used for typesetting if your manuscript is accepted.

If you have been asked to revise the written English in your submission as a condition of publication, you must do so, and you are expected to provide evidence that you have received language editing support. The journal would prefer that you use a professional language editing service and provide a certificate of editing, but a signed letter from a colleague who is a fluent speaker of English is acceptable. Note the journal has arranged a number of discounts for authors using professional language editing services (<https://royalsociety.org/journals/authors/benefits/language-editing/>).

===PREPARING YOUR REVISION IN SCHOLARONE===

-- Ensure that your data access statement meets the requirements

at <https://royalsociety.org/journals/authors/author-guidelines/#data>. You should ensure that you cite the dataset in your reference list. If you have deposited data etc in the Dryad repository, please include both the 'For publication' link and 'For review' link at this stage.

-- If you have uploaded ESM files, please ensure you follow the guidance

at <https://royalsociety.org/journals/authors/author-guidelines/#supplementary-material> to include a suitable title and informative caption. An example of appropriate titling and captioning may be found at https://figshare.com/articles/Table_S2_from_Is_there_a_trade-off_between_peak_performance_and_performance_breadth_across_temperatures_for_aerobic_scope_in_teleost_fishes_/3843624.

Appendix B

Reviewer: 1

Comments to the Author(s)

The authors have improved the manuscript. I have the following minor comments:

1) “Minimal cell rearrangement is not observed over the first 2 hours post culture [57], but appears to occur later as the explants become polarised with respect to germ layer markers [56]. “

This is not understandable. What is ‘minimal’, why minimal?

We have now corrected this error and re-written the sentence to improve clarity

2) “Furthermore, multi-dimensional modelling of plant development has revealed significant insights into the intersections of tissue morphogenesis and patterning [].”

Missing reference

This missing reference has now been added.

Reviewer: 2

Comments to the Author(s)

The authors have adequately addressed my concerns and I now support publication.